# Whole genome sequencing and phylogenetic classification accelerate the implementation of respiratory syncytial virus genomic surveillance in Canada: a pilot study

Ruimin Gao,[1] Cody Buchanan,[1] Kerry Dust,[2] Paul Van Caeseele,[2] Henry Wong,[3] Calvin Sjaarda,[3] Prameet M. Sheth,[3] Agatha N. Jassem,[4] Jessica Minion,[5] Nathalie Bastien[1]

**ABSTRACT**   Whole genome sequencing (WGS) has emerged as a powerful tool to facilitate the study of existing and emerging infectious diseases. WGS-based genomic surveillance provides information on the genetic diversity and tracks the evolution of important viral pathogens, including respiratory syncytial virus (RSV). Multiplex tiling polymerase chain reaction (PCR) assays have been used to facilitate sequencing of a variety of pathogens in support of genomics-based surveillance initiatives. We developed, optimized, and implemented multiplex tiling PCR assays for RSVA and RSVB capable of generating near-complete genomes in the majority of contemporaneous specimens tested. A pilot data set comprising 52 RSVA and 37 RSVB genomes derived from Canadian clinical specimens during the 2022–2023 respiratory virus season was used to perform phylogenetic analyses using both near-complete genome and glycoprotein (G) sequences. Overall, the RSV phylogenetic tree built with whole genomes showed identical lineage clusters as compared to the *G* gene but was more discriminatory. Moreover, the availability of complete genomes enables the identification of a broader range of mutations. For instance, mutations identified in the fusion protein among Canadian isolates tested here, including S377N, K272M, S276N, S211N, S206I, and S209Q, could affect the efficacy of current vaccines or antiviral-based therapeutics. In conclusion, our work reinforces other recent studies demonstrating the utility of multiplex tiling PCR assays to facilitate high-throughput WGS of RSV, which is capable of supporting enhanced genomic surveillance initiatives, as well as the more comprehensive genomic analyses required to inform public health strategies for the development and usage of vaccines and antiviral drugs.

**IMPORTANCE**   We present assays to efficiently sequence genomes of RSVA and RSVB. This enables researchers and public health agencies to acquire high-quality genomic data using rapid and cost-effective approaches. Genomic data-based comparative analysis can be used to conduct surveillance and monitor circulating isolates for efficacy of vaccines and antiviral therapeutics.

**KEYWORDS**   RSV, multiplex tiling PCR, whole genome sequencing, genomic surveillance, phylogeny, vaccine and antiviral drug

Respiratory syncytial virus (RSV), also called human respiratory syncytial virus (hRSV), is a common, contagious airborne viral infection that primarily affects the respiratory tract (1, 2). RSV is a negative-sense enveloped single-stranded non-segmented RNA virus belonging to the *Paramyxoviridae* family, genus *Orthopneumovirus*. This virus has a ~15.2 kb genome that encodes 11 proteins, including 3 surface glycoproteins: the fusion protein (F), attachment glycoprotein (G) and the small hydrophobic protein (SH),

Address correspondence to Ruimin Gao, ruimin.gao@phac-aspc.gc.ca, or Nathalie Bastien, nathalie.bastien@phac-aspc.gc.ca.

The authors declare no conflict of interest.

the RNA-dependent RNA polymerase large protein (L), nucleocapsid (N), phosphoprotein (P), transcriptional regulators (M2-1 and M2-2), matrix (M), and non-structural proteins (NS1 and NS2) (3). The F and G proteins promote the production of protective immune response, and antigenic differences in the G, F, and SH envelope proteins have led to the classification of RSV into two major antigenic groups: RSVA and RSVB. The F protein is responsible for the fusion of the viral envelope with the host cell membrane for the viral entry into the cell and is highly conserved and immunogenic, which is attractive for vaccine development. In contrast, the G protein, which is responsible for cellular attachment, is prone to frequent mutations (3–5).

RSV diagnosis is usually based on clinical symptoms, but laboratory tests such as rapid antigen tests (6, 7) and polymerase chain reaction (PCR) (8) are commonly used diagnostic methods. In recent years, whole genome sequencing (WGS) and its analysis have provided a more in-depth understanding of RSV outbreaks (9–11). By identifying and characterizing viral isolates causing outbreaks, public health authorities and institutions can implement appropriate control measures, such as isolation, contact tracing, and targeted vaccination campaigns (11). WGS-based genomic surveillance involves the systematic monitoring and analysis of viral genomes to understand their genetic diversity, track transmission patterns, and inform public health interventions (12). It has been instrumental in characterizing RSV isolates circulating globally, identifying emerging clades or subclades, and assessing their impact on disease severity and vaccine efficacy. Initially based on sequences derived from the *G* gene, five RSVA and four RSVB clades were identified, named GA1–GA5 and GB1–GB5, respectively. Nextclade (https://clades.nextstrain.org/) has now included both the G_Clade (13) and recently standardized hRSV Genotyping Consensus Consortium (RGCC) lineages for classifications (14, 15).

WGS-based genomic surveillance provides complete genetic profiles on the pathogens and enables researchers to identify specific genetic mutations, including single-nucleotide polymorphisms, insertions, deletions, and rearrangements throughout the viral genomes. Especially since the RNA-dependent replication cycle of RSV is error prone with no proofreading mechanism (16). Analysis of genetic variations and evolutionary patterns informs our understanding of how the virus evolves over time and how new isolates or variants emerge. Furthermore, genomic surveillance helps monitor for genetic changes that may affect the antigenicity of RSV isolates (13). By examining specific genomic regions associated with viral surface proteins (e.g., the G, F, and SH), researchers can identify potential variations that might affect the effectiveness of vaccines or antiviral-based therapeutics or prophylactic monoclonal antibodies (mAbs). This information informs vaccine development strategies and the selection of vaccine isolates (17).

Although there are a number of RSV vaccine candidates in the pipeline (18–20), since late 2023, only two have been approved for use in Canada for adults over 60 years of age, including RSVPreF3 (Arexvy) and RSVpreF (Abrysvo) (20, 21). Both comprise elements of the RSV F protein stabilized in its prefusion conformation (22). Additionally, there are two prophylactic RSV mAbs approved for use in Canada, including palivizumab (SYNAGIS), which is targeted against the F antigenic site II, and nirsevimab (Beyfourtus), which is targeted against the F antigenic site Ø. In Canada, the use of palivizumab has been reserved for at-risk infants and children under 2 years of age largely due to cost, limited efficacy, and the requirement of multi-dose regimen. Conversely, nirsevimab, approved for use in Canada in 2023, requires only a single dose and is being recommended for all newborns (22). Nirsevimab makes use of specifically engineered amino acid (aa) changes (M257Y/S259T/T261E) to the fragment crystallizable region of the immunoglobulin G antibody that results in an extended half-life in serum capable of providing protection with one injection for the entire RSV season (23, 24). Thus, with wider use of both vaccines and mAbs, it is critical to perform both serological and genomics-based surveillance to identify any shift in antigenicity among circulating viruses that could impact RSV vaccines and therapeutics used in Canada (20).

It is worth mentioning that the coronavirus disease 2019 pandemic has significantly influenced research activities across various fields, including those for RSV. Although several WGS-based methods for RSV have been reported previously, their strategies relied upon amplification of many individual PCR products before pooling them for sequencing (25, 26), hybridization to capture probes (27, 28), or metagenomic sequencing (29), which are time-consuming and not ideal for high-throughput routine analysis. In contrast to these methods, the multiplex tiling PCR amplification approach is more amenable to routine and high-throughput sequencing operations. Given recent approvals in Canada and abroad for vaccines targeting RSV, we and others have developed assays targeting RSVA and RSVB to support surveillance initiatives (30–32). To validate the assays described here, and to explore both the genomic diversity of RSV and carriage of important mutations within strains circulating in Canada, we conducted a pilot study comprising 89 representative clinical specimens from four different provinces, including Ontario (ON), Manitoba (MB), British Columbia (BC), and Saskatchewan (SK). Our results demonstrate that the multiplex tiling PCR approach and subsequent genomic analyses have great potential to accelerate large-scale RSV sequencing, diagnosis, and genomic surveillance in Canada. Furthermore, these multiplex tiling PCR assays enable the simultaneous amplification of the complete genome of this virus, allowing for a more comprehensive analysis of its genetic diversity and evolution with enhanced efficiency and sensitivity, which will enable researchers to better understand RSV and inform public health strategies regarding vaccine and antiviral development and usage.

## MATERIALS AND METHODS

### Reference viruses and clinical specimens

Reference RSV isolates used in this study were purchased from Cedarlane and include VR-955 (strain 9320, originally collected in 1977), VR-1803 (strain ATCC-2012-11, originally collected in 2012) and VR-1794 (strain ATCC-2012-10, originally collected in 2012). RSV-positive nasopharyngeal swabs were kindly provided by four provincial laboratories, including the British Columbia Center for Disease Control (BC), Roy Romanow Provincial Laboratory (SK), Cadham Provincial Laboratory (MB), and Kingston Health Sciences Center (ON). All specimens were collected during the 2022–2023 respiratory virus season, with the exception of four specimens from 2016 (SK), one from 2019 (MB), and two from 2021 (BC).

### Development of the multiplex tiling PCR amplification assays

#### *RSVA*

The initial assay was designed against a data set comprising all publicly available complete RSVA genomes ($n$ = 869, accessed 22 November 2022) downloaded from the Bacterial and Viral Bioinformatics Resource Center (BVBRC) (https://www.bv-brc.org/). The genomes were clustered at 99% sequence identity using cd-hit-est (v.4.8.1) (33), resulting in 71 clusters, and the corresponding genomes representing each cluster were aligned using MAFFT (v.7.505) (34). The alignment was reordered such that the first sequence corresponded to the largest cluster with subsequent sequences representing smaller clusters. The reordered alignment was processed using PrimalScheme (v.1.3.2) (30), which was used to develop a multiplex tiling PCR scheme with default settings, except the amplicon size range was iterated in 100 nucleotide (nt) increments from 400 to 500 nt through 1,100–1,200 nt to identify a primer set with the broadest coverage across the genome. The final assay used an amplicon size range of 800–900 nt, and the RSVA isolate, SE01A-0167-V02 (GenBank accession no. MZ515773.1), was selected as the coordinate system for PrimalScheme (30). An additional 2,527 RSVA genomes were acquired from Global Initiative on Sharing All Influenza Data (GISAID) (accessed 24 November 2022) and combined with those from the BVBRC for a combined data set comprising 3,396 genomes. After low-quality (>10% ambiguous bases) and

mislabeled sequences were removed, a total of 3,356 sequences were aligned using MAFFT (v.7.505), and the primer outputs by PrimalScheme were mapped against this alignment in Geneious Prime (v.2023.2.1, Biomatters Ltd). The primers were manually modified (i.e., shifted upstream/downstream, application of degenerate nucleotides, or the development of alternate/supplementary primers) as required to account for any genetic diversity, correct obvious flaws within the priming region, and to ensure that the 5′ and 3′ ends of the genome were adequately captured, such that all coding regions could be recovered.

### RSVB

The assay for RSVB was designed similarly as described above for RSVA, but with some changes to simplify the primer design process. The RSVB design was targeted against more contemporary isolates that have been isolated since 2018, which comprised a total of 2,126 partial (≥8,000 bases) and complete genomes downloaded from BVBRC ($n$ = 437, accessed 2 March 2023) and GISAID ($n$ = 1,692; accessed 2 March 2023). After low-quality (>10% ambiguous bases) and mislabeled sequences were removed, a total of 2,097 sequences were clustered at 99%, 98% and 97% sequence identity using cd-hit-est (v.4.8.1) (33). The set of representative sequences output by cd-hit-est from each sequence identity level was aligned using MAFFT (v.7.505) (34), and a consensus sequence was generated from each data set in Geneious Prime (v.2023.2.1, Biomatters Ltd) using the majority nucleotides at each position. The consensus sequences were used as input for PrimalScheme (v.1.3.2) with default settings, and the RSVB isolates, HRSV/B/Bern/2019 (GenBank accession no. MT107528.1), were used as the coordinate system to develop a multiplex tiling PCR scheme using default settings, except the amplicon size was set to 500 nt. The primer outputs by PrimalScheme were mapped against an alignment comprising the initial data set of 2,097 sequences and manually modified as required to account for any genetic diversity, correct obvious flaws within the priming region, and to ensure that the 5′ and 3′ ends of the genome were adequately captured such that all coding regions could be recovered.

Prior to ordering, the primers from both assays were reassessed *in silico* against their respective data sets to identify any primers that required modification (i.e., shifted up/downstream, application of degenerate nucleotides, or the development of alternate/supplementary primers) to correct for mutations in their extending ends, and/or to account for diversity not captured during the design with PrimalScheme.

### RNA extraction, quantitative real-time PCR, and cDNA synthesis

Viral RNA was extracted from 265 µL of clinical specimens using the Magmax-96 Viral RNA Isolation Kit (cat. no. AMB1836-5, Life Technologies–Invitrogen) as per the manufacturer's protocol, and the samples were processed on the Thermo Scientific KingFisher Flex Purification System. Viral RNA was eluted in 90 µL of Tris elution buffer and either used immediately or stored at −80℃. The presence of RSV and subtypes was confirmed by quantitative real-time PCR (qRT-PCR) using Invitrogen SuperScript III Platinum One-Step qRT-PCR System (cat. no. 11732088) with RSV subtyping primers and probes based on the *L* gene (35) and listed here in Table 1. The temperature cycles were one cycle at 50℃ for 30 min, one cycle at 95℃ for 2 min, 45 cycles at 95℃ for 15 s and 55℃ for 30 s. SuperScript IV Reverse Transcriptase (cat. no. 18090200, Invitrogen) was used to synthesize cDNA from 5 µL of RNA in conjunction with 0.5 µL of 60 µM random hexamers (cat. no. S1330S, New England Biolabs [NEB]) in a final reaction volume of 10 µL. The reverse transcription reaction was incubated at 42℃ for 50 min followed by 70℃ for 10 min.

### Preparation of primer pools and multiplex tiling PCR amplification

For each assay, multiplex primer pools were prepared for each set of primers (i.e., pool 1 and pool 2) by combining equal volumes of the appropriate primers (LabReady, 100 µM

**TABLE 1** Primer/probe sequences based on *L* gene used for identifying RSV subtypes

| Primer/probe names | Sequences (5'–3') |
| --- | --- |
| RSV_Forward | AATACAGCCAAATCTAACCAACTTTACA |
| RSV_Reverse | GCCAAGGAAGCATGCAATAAA |
| RSVA_probe | FAM-TGCTATTGTGCACTAAAG-MGBNFQ |
| RSVB_probe | FAM-CACTATTCCTTACTAAAGATGTC-MGBNFQ |

IDTE, pH 8.0; IDT) and then diluting them to 5 µM prior to use. Two separate 25 µL PCR mixtures, one for each primer pool, were prepared using the Q5 Hot Start High-Fidelity 2× Master Mix (cat. no. M0494L, NEB) as follows: 12.5 µL of Q5 2× Master Mix, 7.5 µL of nuclease-free water, 2 µL of cDNA, and 3 µL of either 5 µM primer pool 1 or pool 2. PCR amplification was carried out on a MiniAmp Plus Thermal Cycler (Applied Biosystems by Thermo Fisher Scientific) with an initial denaturation stage at 98°C for 30 s, followed by 34 cycles at 98°C for 15 s, 62°C for 5 min, and a final hold at 4°C. The PCR products from each reaction were combined then purified using an equal volume of PCRClean DX (cat. no. C-1003-250, Aline Biosciences) as per the manufacturer's protocol. The 1× dsDNA High Sensitivity Kit (cat. no. Q33230, Invitrogen) was used to quantify 1 µL of the purified PCR product on the Qubit Flex fluorometer (Invitrogen) as per the manufacturer's protocol, and the purified PCR products were normalized to 16 ng/µL with 0.01 M Tris in preparation for sequencing.

Each assay was then optimized iteratively against specimens available in-house to identify and correct poorly amplifying or dropout regions either by modulation of the relative primer concentrations or development of additional replacement primers. The finalized primer sequences and corresponding ratios for both assays are listed in Tables 2 and 3.

## Nanopore library preparation and sequencing workflow

A total of 12.5 µL of normalized PCR product (~200 ng) from each sample was used as input to generate barcoded sequencing libraries using the Native Barcoding Kit (cat. no. SQK-NBD114-96, ONT). The protocol was followed exactly with the exception that the NEBNext FFPE DNA Buffer and Repair Mix were not used and replaced with 1.75 µL of Ultra II End-prep Reaction Buffer. The final library, comprising 12 µL of the pooled libraries, 37.5 µL of sequencing buffer, and 25.5 µL of library beads, was loaded onto a FLO-MIN114 R10.4.1 flow cell with a max of 96 specimens and sequenced for 72 h on the MinION Mk1C Sequencing Platform.

## Bioinformatics analysis of sequence data obtained from Nanopore platform

The FAST5 files were basecalled and demultiplexed using Guppy (v.6.5.7) with default settings except specification of dna_r10.4.1_e8.2_400bps_5khz_hac as the basecaller model and SQK-NBD114-96 as the barcode kit with the "--require-bar-code_both_ends" parameter. The basecalled FASTQ files were processed using the Nextflow-enabled viralassembly pipeline (https://github.com/phac-nml/viralassembly), which is a genericized version of the ncov2019-artic-nf pipeline (https://github.com/connor-lab/ncov2019-artic-nf) that automates the ARTIC Network's Field Bioinformatics Toolkit (https://github.com/artic-network/fieldbioinformatics). The viralassembly pipeline is capable of processing any multiplex tiling PCR assay as long as the reference sequence (fasta format) and primer coordinate file (bed format) used to create the assay are provided (File S1 through S6). Briefly, the viralassembly pipeline automates read mapping, primer trimming, variant calling, and consensus sequence generation. Default settings were used, except Medaka (https://github.com/nanoporetech/medaka) was selected as the variant caller in conjunction with the r1041_e82_400bps_hac_v4.2.0 model, and the maximum read length to keep was set to 1,500 nt for the RSVA assay and 1,000 nt for the RSVB assay.

**TABLE 2** RSVA multiplex PCR primer pools

| Pool | Name | Sequences | Ratio |
|---|---|---|---|
| Pool 1 | hRSVA_99_800-900_v3_1_LEFT | TGTTATTACAAGTAGTGATATTTGCCCY | 5× |
| | hRSVA_99_800-900_v3_1_RIGHT | TTGCCCCATCTTTCATCTTATRT | 5× |
| | hRSVA_99_800-900_v3_3_LEFT | TGAAATGAAACGTTATAAAGGYTTAYTACC | 5× |
| | hRSVA_99_800-900_v3_3_RIGHT | AATTGGGCTTGTTCCCTRCAGT | 5× |
| | hRSVA_99_800-900_v3_5_LEFT | GAAGCTATGGCAAGACTCAGGA | 1× |
| | hRSVA_99_800-900_v3_5_RIGHT | TCAAGTGTGTTCAGATCTTTATTTCTGA | 1× |
| | hRSVA_99_800-900_v3_7_LEFT | AAGCAAATTCTGGCCTTACTTTAC | 1× |
| | hRSVA_99_800-900_v3_7_RIGHT | GTTGGATTGTTGCTGCATATGCT | 1× |
| | hRSVA_99_800-900_v3_9_LEFT | CAAACCTCAAACCACAAAACCAAAR | 2× |
| | hRSVA_99_800-900_v3_9_RIGHT | GCRATTGCAGATCCAACACCTA | 2× |
| | hRSVA_99_800-900_v3_11_LEFT | TGTAACTACACCTGTAAGCACTTATATGT | 1× |
| | hRSVA_99_800-900_v3_11_RIGHT | GTAGTTATCATGATATTTGTGGTGGATTTAC | 1× |
| | hRSVA_99_800-900_v3_13_LEFT | ATAAGTGGAGCTGCAGAGTTGG | 1× |
| | hRSVA_99_800-900_v3_13_RIGHT | AGGACCATTGAATATGTAACTTCCTAARG | 1× |
| | hRSVA_99_800-900_v3_15_LEFT | CTATGCTATATTGAATAAACTGGGGCT | 3× |
| | hRSVA_99_800-900_v3_15_RIGHT | AGGTTRTTGTCACCTGCAAGYT | 3× |
| | hRSVA_99_800-900_v3_17_LEFT | AGATGGTTAACTTACTATAAACTAAACAC | 2× |
| | hRSVA_99_800-900_v3_17_RIGHT | TCCATAGTTTTTGACACCACCCTT | 2× |
| | hRSVA_99_800-900_v3_19_LEFT | AGTATAAAGAAAGTCCTAAGAGTGGGAC | 1× |
| | hRSVA_99_800-900_v3_19_LEFT_alt1 | TAGTATAAAGAAAGTCCTAAGAGTRGG | 1× |
| | hRSVA_99_800-900_v3_19_RIGHT | GCAGAAGTCTTTTCCAGTATGTTAGT | 1× |
| | hRSVA_99_800-900_v3_21_LEFT | ACTATAGCTAGTGGCATAATCATAGARAA | 2× |
| | hRSVA_99_800-900_v3_21_RIGHT | TCCCTCTCCCCAATCTTTTTCAA | 2× |
| | hRSVA_99_800-900_v3_23_LEFT | AATTACAACAAATTATATCATCCYACACC | 3× |
| | hRSVA_99_800-900_v3_23_RIGHT | CATCTTGAGCATGATATTTTACTATTAAYGTAC | 3× |
| | hRSVA_99_800-900_v3_25_LEFT | AGAGTGTTGTTAGTGGAGATATACTATC | 5× |
| | hRSVA_99_800-900_v3_25_RIGHT | ACGAGAAAAAAGTGTCAAAAACTAATRTC | 5× |
| | hRSVA_99_800-900_v3_25_RIGHT_alt2 | AATATACATATAAACCAATTAGATTTGGATTTAA | 5× |
| Pool 2 | hRSVA_99_800-900_v3_0_LEFT | CGAAAAAATGCGTACAACAAACTT | 2× |
| | hRSVA_99_800-900_v3_0_LEFT_alt1 | GGGCAAATAAGAATTTGATAAGTACCA | 2× |
| | hRSVA_99_800-900_v3_0_RIGHT | ACTTTGTGCAATAGTTTCATTTCATAGTT | 2× |
| | hRSVA_99_800-900_v3_2_LEFT | CCCATAATATACAAGTATGATCTCAATCCRT | 1× |
| | hRSVA_99_800-900_v3_2_RIGHT | CCACCTCTGGTAGAAGATTGTGC | 1× |
| | hRSVA_99_800-900_v3_4_LEFT | TGACAGCAGAAGAACTAGAGGC | 1× |
| | hRSVA_99_800-900_v3_4_RIGHT | TCAGAAATCTTCAAGTGATAGATCATTRTC | 1× |
| | hRSVA_99_800-900_v3_6_LEFT | GATCTCACTATGAAAACACTCAAYCC | 1× |
| | hRSVA_99_800-900_v3_6_RIGHT | TTGACTCGAGCTCTTGGTARY | 1× |
| | hRSVA_99_800-900_v3_8_LEFT | CCAGATCAAGAACACAACCCCAA | 1× |
| | hRSVA_99_800-900_v3_8_RIGHT | GCAACTCCATTGTTATTTGCCCC | 1× |
| | hRSVA_99_800-900_v3_10_LEFT | CAAAGCACACCAGCAGCMAA | 2× |
| | hRSVA_99_800-900_v3_10_RIGHT | TTGAATATGTCAATGTTGCAGAGATTTAC | 2× |
| | hRSVA_99_800-900_v3_12_LEFT | TCCCYTCTGATGAATTTGATGCA | 1× |
| | hRSVA_99_800-900_v3_12_RIGHT | TGAGTTCAGTRAGGAGTTTGCTCA | 1× |
| | hRSVA_99_800-900_v3_14_LEFT | TGATACTACCTGACAAATATCCTTGTAG | 1× |
| | hRSVA_99_800-900_v3_14_RIGHT | TTGTTTTGWGGATTGATTTTTGYCTG | 1× |
| | hRSVA_99_800-900_v3_14_RIGHT_alt1 | TATTAACCATGATGGAGGATGTTGC | 1× |
| | hRSVA_99_800-900_v3_16_LEFT | TGCTCAACAACATCACAGATGC | 1× |
| | hRSVA_99_800-900_v3_16_RIGHT | TCTGCTAATATTTGAACTTGTCTGAACA | 1× |
| | hRSVA_99_800-900_v3_18_LEFT | TCCTGGTTACATTTAACTATTCCTCATG | 1× |
| | hRSVA_99_800-900_v3_18_RIGHT | AGGAAATCAGGAGTTCTTCTATAGAAACT | 1× |
| | hRSVA_99_800-900_v3_20_LEFT | AACCTACATATCCTCAYGGGCT | 2× |
| | hRSVA_99_800-900_v3_20_RIGHT | AGCTAAGGCCAAARCTTATACAGT | 2× |

*(Continued on next page)*

**TABLE 2** RSVA multiplex PCR primer pools (*Continued*)

| Pool | Name | Sequences | Ratio |
|------|------|-----------|-------|
| | hRSVA_99_800-900_v3_22_LEFT | TCTMATGTTAATTCTAATTTAATATTGGCRCA | 5× |
| | hRSVA_99_800-900_v3_22_RIGHT | TCTTGTYTGCTGTAATTGGTTCTAATC | 5× |
| | hRSVA_99_800-900_v3_24_LEFT | GTTCTACAGGTTGTAAAATTAGTATAGAGT | 3× |
| | hRSVA_99_800-900_v3_24_RIGHT | CTGTTGATCTGAAATTTAAAACATGRTTGAACC | 3× |

For the purposes of this study, a genome was considered to be complete if it encompassed the first codon of the NS1 protein and the last codon of the L protein (15). The open reading frames (ORFs) encoding the RSV genes were identified using Geneious Prime (v.2023.2.1, Biomatters Ltd) and characterized as a means to detect and correct any sequencing errors or to validate legitimate biological mutations that would result in the formation of a truncated protein.

## RSVA and RSVB data sets

In order to contextualize our sequences within the broader population structure of RSV, 124 RSVA and 83 RSVB lineage exemplar reference sequences (https://github.com/rsv-lineages) were downloaded from the National Center for Biotechnology Information (NCBI). They were combined with the 52 RSVA and 40 RSVB sequences generated here (37 Canadian + 3 American Type Culture Collection [ATCC] isolates), resulting in data sets comprising 176 RSVA and 123 RSVB sequences, respectively (File S7).

## Characterization of RSVA and RSVB sequences using Nextclade

The RSVA and RSVB data sets were processed using Nextclade's web portal on 9 August 2024 (https://clades.nextstrain.org/) and analyzed using the RSVA module against the reference isolate hRSV/A/England/397/2017 (EPI_ISL_412866) and the RSVB module against the reference isolate hRSV/B/Australia/VIC-RCH056/2019 (EPI_ISL_1653999), respectively. The RGCC lineage assignments and G_Clade typing nomenclatures (15, 36), as well as quality metrics and genomic features including the genome length, breadth of coverage, and the presence of variants including nt and aa substitutions, insertions, deletions and frameshifts, are described in File S7. Amino acid changes identified in the F protein among the 52 RSVA and 37 RSVB sequences were tabulated and used to generate heatmaps depicting their carriage across the data sets. The heatmaps were generated using a custom R script implemented in RStudio 2023.12.1 Build 402 running R (v.4.3.2, 31 October 2023) with the following packages: ComplexHeatmap (v.2.10.0), tidyverse (v.2.0.0), magrittr (v.2.0.3), grid (v.4.1.2), and vegan (v.2.6.4). The R code used to generate the heatmaps is contained in File S8.

## Whole genome phylogenetic analysis

The RSVA and RSVB data sets were aligned separately with MAFFT (v.7.520) using default settings (34), then processed with FastTree to infer approximate maximum-likelihood phylogenetic trees with bootstrapping using default settings and the generalized time reversible (GTR) model (37, 38). The phylogenetic trees were visualized using the Interactive Tree Of Life (iTOL) tool (39) and annotated with the corresponding sequence metadata captured in File S7.

## *Glycoprotein* (*G*) gene phylogenetic analysis

The G ORF for all RSVA and RSVB samples was identified using the "Find ORFs" function in Geneious Prime (v.2023.2.1, Biomatters Ltd), and the corresponding coding sequences were extracted and subjected to BLAST analysis to confirm they were in-frame and intact. The G sequences from RSVA and RSVB were each aligned separately using MAFFT (v.7.520) with default settings (34), then maximum-likelihood trees were inferred using FastTree with default settings and the GTR model (37, 38). The phylogenetic trees were

**TABLE 3** RSVB multiplex PCR primer pools

| Pool | Name | Sequences | Ratio |
|---|---|---|---|
| Pool 1 | hRSVB_2018-2023_500_1_LEFT | CTAGACTCCGTCACGCGAAAAA | 2× |
| | hRSVB_2018-2023_500_1_LEFT_alt1 | AAATGCGTACTACAAACTTGCACACTC | 2× |
| | hRSVB_2018-2023_500_1_RIGHT | GGAGATCAAGCCCAAGTAAATCAGA | 2× |
| | hRSVB_2018-2023_500_3_LEFT | TCAATTATGAGATGAAGCTATTGCACA | 1× |
| | hRSVB_2018-2023_500_3_RIGHT | TGCATCTTCAGTGATTAATAGCATACC | 1× |
| | hRSVB_2018-2023_500_5_LEFT_alt1 | CCAGACTGTGGGATGATAATACTGTG | 1× |
| | hRSVB_2018-2023_500_5_RIGHT | TTGCCTAGGACCACACTTGAGA | 1× |
| | hRSVB_2018-2023_500_7_LEFT_alt1 | CAAGTTTGCATCATCCAAAGATCCTAA | 1× |
| | hRSVB_2018-2023_500_7_RIGHT_alt1 | TCTTGCCATGGCCTCTAACCTAT | 1× |
| | hRSVB_2018-2023_500_7_RIGHT_alt2 | GCCTCTAACCTATCATTGGTCAT | 1× |
| | hRSVB_2018-2023_500_9_LEFT | GGAAACATACGTGAACAAGCTTCA | 1× |
| | hRSVB_2018-2023_500_9_RIGHT_alt1 | ACACTGATTGATCTTAGATAGGTTGGTATT | 1× |
| | hRSVB_2018-2023_500_11_LEFT_alt1 | TCCTCAACTGCACACTATATCTAAACATC | 1× |
| | hRSVB_2018-2023_500_11_RIGHT | TGTGATGCCATGACTCTGTGAG | 1× |
| | hRSVB_2018-2023_500_13_LEFT | CACAAAGTTACACTAACAACTGTCACA | 10× |
| | hRSVB_2018-2023_500_13_RIGHT_alt1 | TCTGTAGTCTTGGGGGGTTGGTTT | 10× |
| | hRSVB_2018-2023_500_15_LEFT_alt1 | CTGGGGCAAATAACCATGGAGY | 1× |
| | hRSVB_2018-2023_500_15_RIGHT_alt1 | TGTTCACTTCTCCTTCAAGGTG | 1× |
| | hRSVB_2018-2023_500_17_LEFT | TCAATGATATGCCTATAACAAATGATCAGA | 5× |
| | hRSVB_2018-2023_500_17_RIGHT_alt1 | TCCACGATTTTTGTTGGATGC | 5× |
| | hRSVB_2018-2023_500_19_LEFT | TCGTAGATCCGATGAATTATTACATAATGT | 1× |
| | hRSVB_2018-2023_500_19_RIGHT | TGACTGTAGTGGCATCTTCTACC | 1× |
| | hRSVB_2018-2023_500_21_LEFT | CAATACTGTTATATCATACATCGAGAGCA | 1× |
| | hRSVB_2018-2023_500_21_RIGHT_alt1 | GGATCCATTTTGTCCCATAACTTTATTRAG | 1× |
| | hRSVB_2018-2023_500_23_LEFT | ATAAAAGCATGTCCTCGTCTGAAC | 2× |
| | hRSVB_2018-2023_500_23_RIGHT | AGCCCTTTATGATAAACAATGCAACC | 2× |
| | hRSVB_2018-2023_500_25_LEFT_alt1 | TCACAGATGCAGCTATTAAGGC | 1× |
| | hRSVB_2018-2023_500_25_RIGHT | ACTCACGATAGAACCGCAATCC | 1× |
| | hRSVB_2018-2023_500_27_LEFT | TGCAACCAGGTATGTTTAGGCAA | 1× |
| | hRSVB_2018-2023_500_27_RIGHT | AATGATATGGCTTCAATGGTCCAC | 1× |
| | hRSVB_2018-2023_500_29_LEFT_alt1 | AGGTCCATGGATAAATACAATACTTGATGA | 2× |
| | hRSVB_2018-2023_500_29_RIGHT | GCCTGTGGATCCCTCATCAATG | 2× |
| | hRSVB_2018-2023_500_31_LEFT | AAAAACATCAGCGATAGATACAACTGA | 1× |
| | hRSVB_2018-2023_500_31_RIGHT_alt1 | ATGACAGTCCAAGTGTTCCAGTAC | 1× |
| | hRSVB_2018-2023_500_31_RIGHT_alt2 | CATATGACAGTCCAAGTGTTCC | 1× |
| | hRSVB_2018-2023_500_33_LEFT | ACATTTGATGAAACCTCCTATATTTACAGG | 2× |
| | hRSVB_2018-2023_500_33_RIGHT | TGACTTTTTGTTCTAGGAAAACTTTAGACA | 2× |
| | hRSVB_2018-2023_500_35_LEFT_alt1 | CTGCTAACAAAACAAATAAGGATTGCTAA | 1× |
| | hRSVB_2018-2023_500_35_RIGHT_alt1 | ACCAGCTCCTTCACCTATGAATG | 1× |
| | hRSVB_2018-2023_500_37_LEFT | TGGAGTAAGCATGTAAGAAAGTGCA | 1× |
| | hRSVB_2018-2023_500_37_RIGHT_alt1 | TCTAAAGTTTAAAACATGATCCAGCCAT | 1× |
| Pool 2 | hRSVB_2018-2023_500_2_LEFT_alt1 | TGCTCTCAATTAAATGGTCTAATAGATGATAA | 1× |
| | hRSVB_2018-2023_500_2_RIGHT_alt1 | GCATAGGGAATGTGCCATATTTTGTA | 1× |
| | hRSVB_2018-2023_500_4_LEFT_alt1 | ACACTATTCAACGTAGTACAGGAGA | 1× |
| | hRSVB_2018-2023_500_4_RIGHT | CATTGTTTGCCCTCCTAATTACTGC | 1× |
| | hRSVB_2018-2023_500_6_LEFT_alt1 | CAGAAGTTGGGAGGAGAAGC | 1× |
| | hRSVB_2018-2023_500_6_RIGHT | TGTTGGTGCCAGATGTTATCGG | 1× |
| | hRSVB_2018-2023_500_8_LEFT_alt1 | CTCGTGACGGAATAAGAGATGC | 1× |
| | hRSVB_2018-2023_500_8_RIGHT | TGATGCGGGATCATCATCTTTTTC | 1× |
| | hRSVB_2018-2023_500_10_LEFT | ACCCCACTCATGAGATCATTGC | 1× |
| | hRSVB_2018-2023_500_10_RIGHT | GCAGACAATGGCTGGAAGTGAT | 1× |
| | hRSVB_2018-2023_500_12_LEFT_alt1 | TCGACACATAGTGTTCTCCCATTAT | 10× |

**TABLE 3** RSVB multiplex PCR primer pools (*Continued*)

| Pool | Name | Sequences | Ratio |
|------|------|-----------|-------|
| | hRSVB_2018-2023_500_12_RIGHT_alt1 | GGCTAACCCTTTCTGGTGAGACT | 10× |
| | hRSVB_2018-2023_500_14_LEFT_alt1 | ACCACAAACAAAAGAGACYCYA | 10× |
| | hRSVB_2018-2023_500_14_RIGHT_alt1 | CCTCAGTTATGTTCTGACTTGAGGY | 5× |
| | hRSVB_2018-2023_500_16_LEFT_alt1 | AGGAAACGAAGATTTCTGGGCTT | 1× |
| | hRSVB_2018-2023_500_16_RIGHT_alt1 | AGCTGTACAACATATGCAAGGAC | 1× |
| | hRSVB_2018-2023_500_18_LEFT_alt1 | CAAAAACAGACATAAGCAGCTCAG | 1× |
| | hRSVB_2018-2023_500_18_RIGHT | TGATTCCACTTAGTTGGTCTTTGCT | 1× |
| | hRSVB_2018-2023_500_20_LEFT | TCACAAAACTAACAGCTGGGGC | 1× |
| | hRSVB_2018-2023_500_20_RIGHT | TTGTCTTCTTCAGCACGTCTGC | 1× |
| | hRSVB_2018-2023_500_22_LEFT | ACATCTTAACATCCCTGAAGATATATATACAGT | 2× |
| | hRSVB_2018-2023_500_22_RIGHT_alt1 | AGTTTATTCAAGATGGCGTACACYT | 2× |
| | hRSVB_2018-2023_500_24_LEFT_alt1 | TCAAATGAGGTAAAAAGTCATGGGT | 2× |
| | hRSVB_2018-2023_500_24_RIGHT | ACCATTTATGATATTATCAGACACTGTCTT | 2× |
| | hRSVB_2018-2023_500_26_LEFT | GCTATTGTCCTACCTCTAAGATGGT | 1× |
| | hRSVB_2018-2023_500_26_RIGHT_alt1 | TGTCAAACTCTCAGGGAAGAATTG | 1× |
| | hRSVB_2018-2023_500_28_LEFT_alt1 | TGAAGTTGATGAACAAAGTGGGTTA | 2× |
| | hRSVB_2018-2023_500_28_RIGHT_alt1 | ACTGCATAATAAGCTTTCTCCTCTGTA | 2× |
| | hRSVB_2018-2023_500_30_LEFT_alt1 | AGCTCCAGGATCTTCCAGAYGAT | 2× |
| | hRSVB_2018-2023_500_30_RIGHT | TCTCTTTTGTCTTTGTTACAATCTAGTGG | 2× |
| | hRSVB_2018-2023_500_32_LEFT_alt1 | CCAAATAGATTTATTAGCAAAATTAGACTGG | 2× |
| | hRSVB_2018-2023_500_32_RIGHT_alt1 | TTGGGTTAAACTTATTTTATCTGGTAGGAAC | 2× |
| | hRSVB_2018-2023_500_34_LEFT_alt1 | GCAAAATTAGAATGTGATATGAACACTTCAGA | 2× |
| | hRSVB_2018-2023_500_34_RIGHT | ACAGGGATTAATGATACATTTTCTAAAGCT | 2× |
| | hRSVB_2018-2023_500_36_LEFT_alt1 | CCTTGGCATCATGTCAATAGATTTAACTT | 1× |
| | hRSVB_2018-2023_500_36_RIGHT | TGAAATCAATATCATCTTGAGCATGGT | 1× |
| | hRSVB_2018-2023_500_38_LEFT | ACTTCATTGTCAAAATTGAAGAGTGTAGT | 1× |
| | hRSVB_2018-2023_500_38_RIGHT | CATGCCGGCCACGAGAAAAA | 2× |
| | hRSVB_2018-2023_500_38_RIGHT_alt1 | AATGTCTCGTTGTGTTGTAAATGCACATR | 2× |

visualized using the iTOL tool (39) and annotated with the corresponding RGCC lineage and G_Clade assignments from Nextclade, as well as the sample collection year and location (File S7).

## Co-phylogeny analysis

For both RSVA and RSVB, a custom R script was used to construct a co-phylogeny comprising the tree files generated from the whole genome and *G* gene coding sequences, and annotated with the corresponding RGCC lineage for each sample. The script was implemented as described above using the following packages managed with pacman (v.0.5.1): ape (v.5.7.1), biocmanager (v.1.30.22), tidyverse (v.2.0.0), ggtree (v.3.10.1), phangorn (v.2.11.1), treeio (v.1.26.0), phytools (v.2.1.1), viridis, here (v.1.0.1), and scico (v.1.5.0). The R code used to generate the co-phylogeny tree is contained in File S8.

## Identification of mutations linked to phenotypic or epidemiological traits using RSVsurver

The RSVsurver, developed by Singapore's Agency for Science, Technology and Research Bioinformatics Institute and enabled by GISAID (https://rsvsurver.bii.a-star.edu.sg/faq.html), was used to screen the RSV sequences against their curated database for the presence of mutations that may be linked to important phenotypic and epidemiological traits. The RSVA and RSVB data sets were uploaded to the RSVsurver, and each sequence was compared against the reference isolates hRSV/A/England/397/2017 and hRSV/B/Australia/VIC-RCH056/2019, respectively.

## RESULTS

### Multiplex PCR-based amplification using tiling scheme of RSV and complete genome sequences collection

For RSVA, PrimalScheme generated an assay comprising 24 overlapping amplicons with 0 gaps accounting for 95.5% of the genome. Additional primer sets were manually designed to generate amplicons capturing regions closer to the 5′ and 3′ ends of the genome extending coverage of the assay to >99% of the genome. For RSVB, Primal-Scheme generated an assay comprising 38 overlapping regions with 0 gaps accounting for 99.6% of the genome. The finalized primer sequences and corresponding ratios for both assays are listed in Tables 2 and 3. Among the in-house 52 RSVA and 37 RSVB clinical specimens, 3 ATCC RSVB stains, and Nextclade curated NCBI 124 RSVA and 83 RSVB reference genomes, the antigenic group, subtypes, genome length, identified mutations, and GISAID IDs are summarized in File S7. The near-complete genome length ranged from 14,994 to 15,225 nt.

### Whole genome phylogenetic analysis of 52 Canadian clinical RSVA isolates

The RSVA data set ($n$ = 176; 52 in-house + 124 reference) was demarcated into nine clades using the G_Clade scheme developed by Goya et al. (36) (File S7; Fig. 1). The majority of sequences were classified as GA.2.3.5 and largely represent more contemporaneous isolates collected since 2007, including all Canadian isolates sequenced in this study (Fig. 1). In contrast, the new scheme proposed by the hRSV RGCC and implemented in Nextclade, was more discriminatory and demarcated the same data set into 25 lineages that more closely reflect the structure of the phylogenetic tree (Fig. 1, represented by colored range with the tree nodes). Overall, lineage assignment was well supported by the phylogenetic tree, though not all sequences from lineages A.D and A.D.5 clustered cohesively. However, this was also apparent in the RSVA maximum-likelihood tree described in Goyal et al. (15) and the publicly available Nextclade RSVA phylogeny.

Isolates derived from clinical specimens characterized in this study, collected between 2016 and 2023, were assigned to nine lineages, all descending from A.D, with the plurality of sequences (25 out of 52) assigned to lineage A.D.5.2 (Fig. 1). The two oldest isolates, collected from SK in 2016, were assigned to lineages A.D and A.D.2.2, respectively, and clustered with reference sequences collected between 2012 and 2016 (blue stars, Fig. 1). Based on available data, lineage A.D.2.2 was prevalent among isolates characterized in 2016, but detections have declined since then (15). Interestingly, the remaining Canadian isolates, collected during the 2022–2023 respiratory virus season and one from 2021, were assigned to multiple lineages, indicating that a diverse array of viruses can be co-circulating simultaneously. With the exception of A.D.5.2 and A.D.1, which represent the most common ($n$ = 25) and second most common ($n$ = 8) lineages detected among the Canadian isolates, a lineage tended to be dominated by sequences from a single province (excluding reference sequences) (Fig. 1).

### RSVA G gene-based phylogenetic analyses and its co-phylogeny analysis with whole genome-based tree

The structure of the phylogenetic tree derived from the complete coding sequence of the *G* gene (Fig. 2A) was largely congruent with the phylogeny derived from the whole genome sequence (Fig. 1). Overall, the lineage designations corresponded to well-defined groups in both phylogenies, though the use of the whole genome sequence showed higher bootstrap values and was better able to resolve the phylogenetic structure within certain sublineages, such as A.D.5.2, and differentiated closely related sequences that were indistinguishable based on the *glycoprotein G* gene sequence (Fig. 1 and 2A). For instance, within the A.D.5.2 lineage, four isolates (RV00279 [CA-SK 2023], RV00288 [CA-SK 2023], RV00290 [CA-SK 2023], and RV00292 [CA-SK 2023]) (green stars,

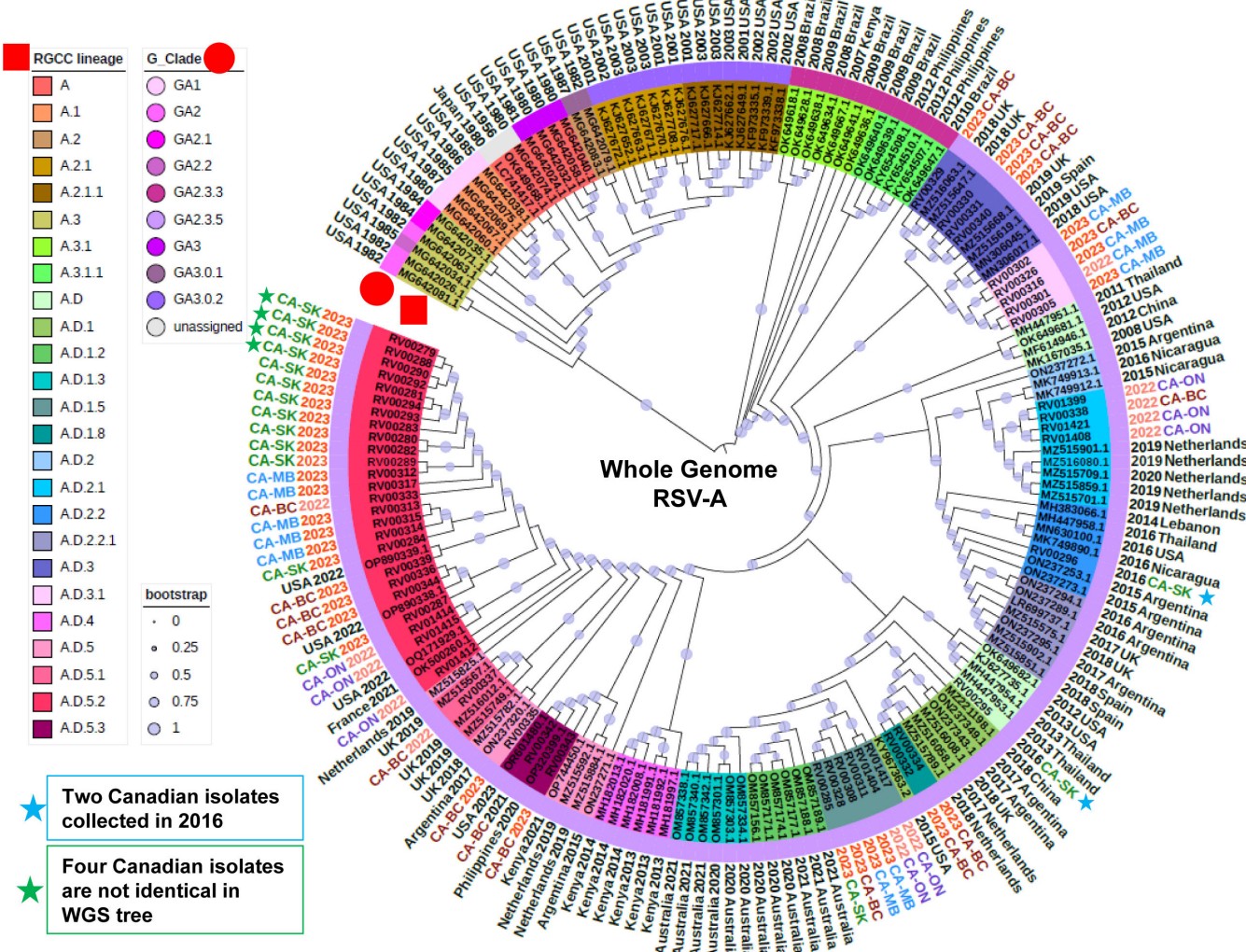

**FIG 1** A phylogenetic tree comprising the 176 RSVA genomes used in this study, including 52 sequences derived from Canadian isolates and 124 Nextclade reference sequences. hRSV Genotyping Consensus Consortium (RGCC) lineages are indicated as colored highlights over the isolate identifier and further annotated with a red square, while the G_Clade classifications are indicated as a color strip above the isolate identifier and further annotated with a red circle. Collection years 2022 and 2023 are indicated with light red and red texts, respectively, while all other collection years are indicated with black text. The Canadian isolates were primarily collected between 2022 and 2023 from four provinces, including British Columbia (BC), Manitoba (MB), Ontario (ON), and Saskatchewan (SK), with corresponding text colors red, blue, purple, and green, respectively. Bootstrap values are annotated as light purple dots on each branch of the tree.

Fig. 2A) were indistinguishable using the *G* gene sequence, but at the whole genome sequence level, could be resolved into two pairs of identical sequences: RV00279 (CA-SK 2023) and RV00288 (CA-SK 2023), as well as RV00290 (CA-SK 2023) and RV00292 (CA-SK 2023) (green stars, Fig. 1).

## Whole genome phylogenetic analysis of 37 Canadian clinical RSVB isolates

Using the new RSVB scheme developed by the RGCC, the RSVB data set (*n* = 123; 37 Canadian + 86 references) was demarcated into 15 lineages that largely reflect the structure of the phylogenetic tree (Fig. 3). Comparatively, using the G_Clade scheme developed by Goya et al. (36), the data set was characterized across seven clades with the majority of sequences classified as GB5.05a, encompassing isolates collected since 2013, including all Canadian isolates sequenced in this study (Fig. 3; File S7). One sequence from each of the B (KP856965) and B.D (MH594451) lineages did not cluster cohesively with other sequences from their respective lineages, but this is also reflected in the

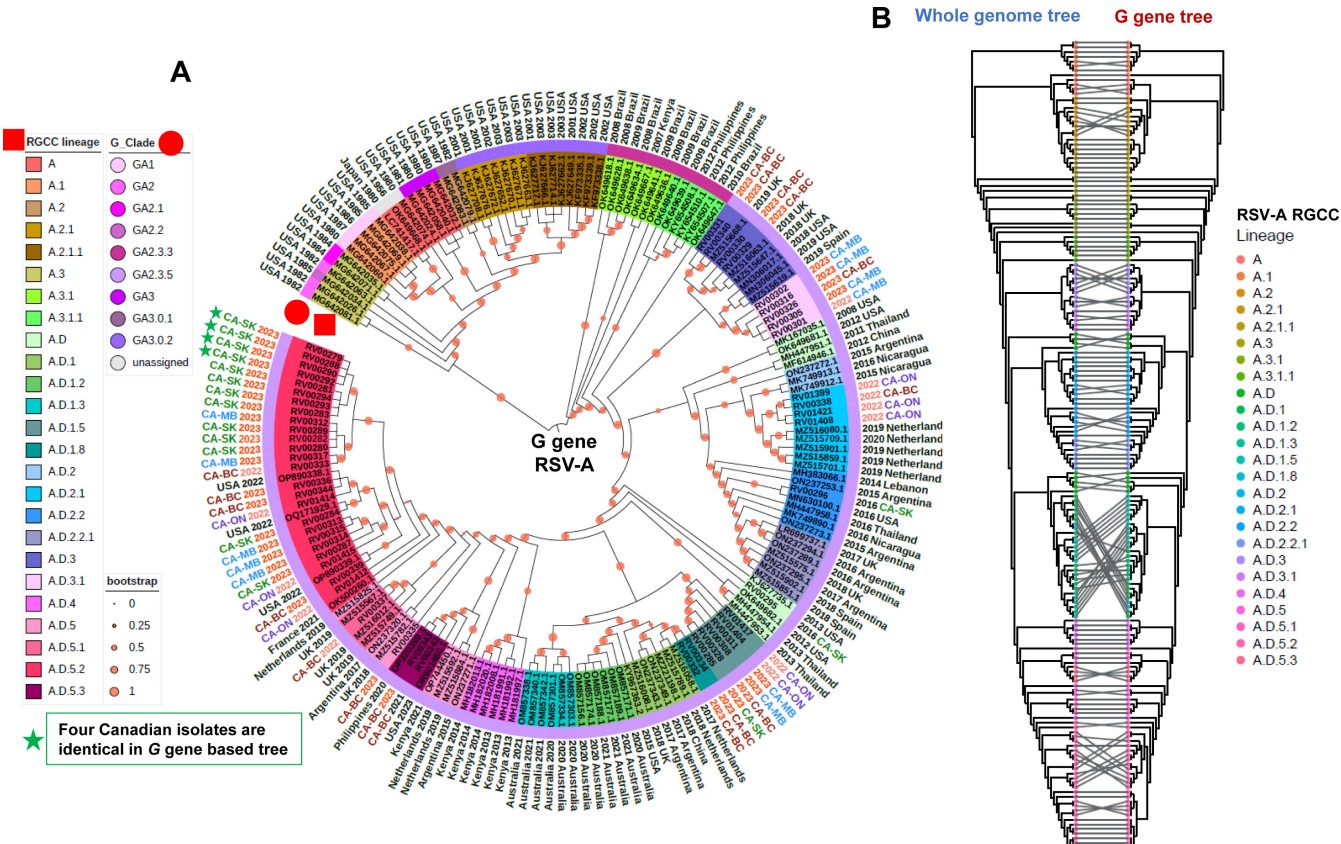

**FIG 2** A phylogenetic tree derived from the *G* gene sequence and a co-phylogeny comparing differences in clustering using the whole genome sequence versus only the corresponding *G* gene sequence based on the 176 RSVA genomes used in this study. (A) A phylogenetic tree based on the *G* gene sequences. The RSV RGCC lineages are indicated as colored highlights over the isolate identifier and further annotated with a red square, while the G_Clade classifications are indicated as a color strip above the isolate identifier and further annotated with a red circle. Collection years 2022 and 2023 are indicated with light red and red texts, respectively, while all other collection years are indicated with black text. The Canadian isolates were primarily collected between 2022 and 2023 from four provinces, including BC, MB, ON, and SK, with corresponding text colors red, blue, purple, and green, respectively. Bootstrap values are annotated as red dots on each branch of the tree. (B) A co-phylogeny comparing differences in clustering using the whole genome versus only the corresponding *G* gene sequence. The corresponding RGCC lineage is annotated at each tip for both phylogenetic trees.

RSVB maximum-likelihood tree described in Goya et al. (15) and the publicly available Nextclade RSVB phylogeny. Isolates derived from the Canadian clinical specimens were demarcated into three lineages with the majority (30 out of 37) assigned to B.D.E.1, which included those from the 2022–2023 respiratory virus season from all provinces, as well as one collected in 2021 from BC. The two isolates collected in 2016 from SK were assigned to B.D.4.1 and clustered with reference sequences collected between 2013 and 2015, while the remaining three isolates, two collected from MB in 2019 and 2023 and one collected from BC in 2022, were assigned to B.D.4.1.1, and were more distantly related to their corresponding NCBI reference isolates (Fig. 3). The reference viruses VR-955 (collected in 1977), VR-1794 (collected in 2012), and VR-1803 (collected in 2012) were assigned to lineages B, B.D, and B.D.4, respectively, and clustered with appropriately contemporaneous reference sequences.

## RSVB *G* gene-based phylogenetic analyses and its co-phylogeny analysis with whole genome-based tree

Similar to RSVA, the structure of the phylogenetic trees derived for RSVB from the complete coding sequences of the *G* gene (Fig. 4A) and whole genome sequence (Fig. 3) was largely congruent (Fig. 4B). Overall, the lineage designations corresponded

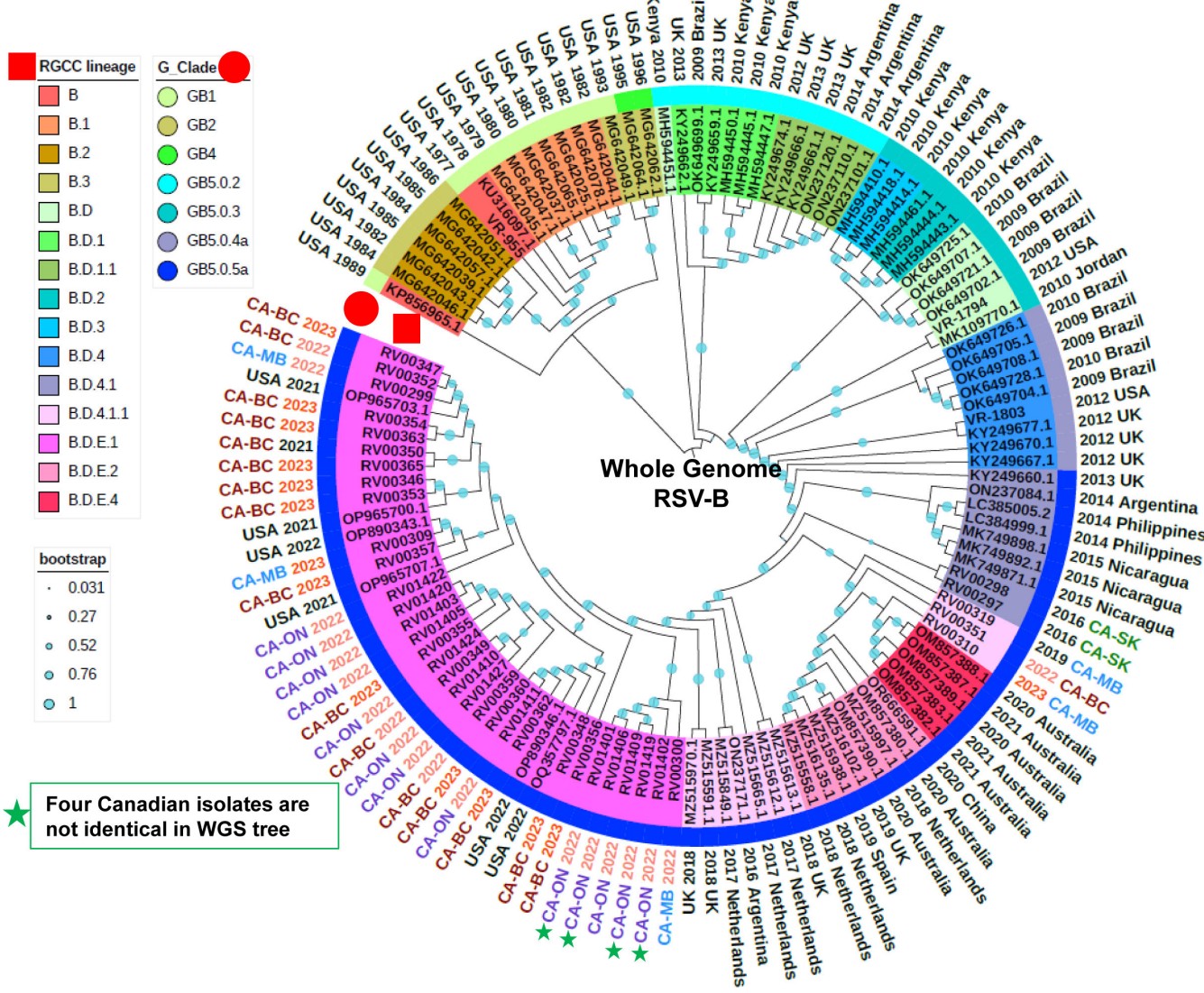

**FIG 3** A phylogenetic tree comprising the 123 RSVB genomes used in this study, including 37 sequences derived from Canadian isolates, 3 ATCC isolates, and 86 Nextclade reference sequences. The RSV RGCC lineages are indicated as colored highlights over the isolate identifier and further annotated with a red square, while the G_Clade classifications are indicated as a color strip above the isolate identifier and further annotated with a red circle. Collection years 2022 and 2023 are indicated with light red and red text, respectively, while all other collection years are indicated with black text. The Canadian isolates were primarily collected between 2022 and 2023 from four provinces, including BC, MB, ON, and SK, with corresponding text colors red, blue, purple, and green, respectively. Bootstrap values are annotated as cyan dots on each branch of the tree.

to well-defined groups in both phylogenies, though the use of the whole genome sequences was better able to resolve the phylogenetic structure within certain sublineages, such as B.D.E.1 and differentiated closely related sequences that were indistinguishable based on the *G* gene sequence (Fig. 3 and 4A). For instance, in the B.D.E.1 lineage, sequences from four isolates, including RV01401 (CA-ON 2022), RV01402 (CA-ON 2022), RV01409 (CA-ON 2022), and RV01419 (CA-ON 2022) (green stars, Fig. 4A) were indistinguishable, but at the whole genome sequence level, could be further resolved into separate clusters (green stars, Fig. 3 and 4A).

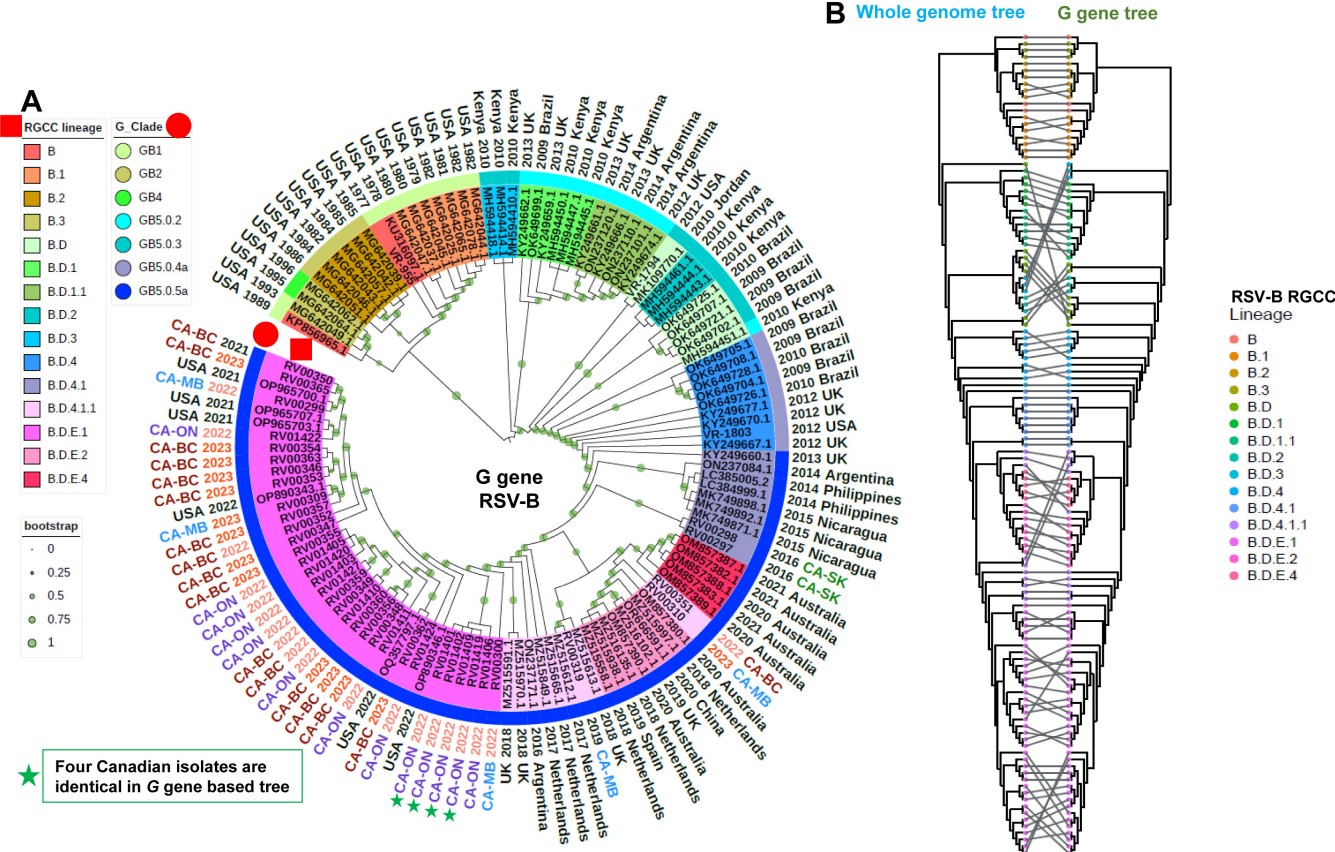

**FIG 4** A phylogenetic tree derived from the *G* gene sequence and a co-phylogeny comparing differences in clustering using the whole genome sequence versus only the corresponding *G* gene sequence based on the 123 RSVB genomes used in this study. (A) A phylogenetic tree based on the *G* gene sequences. The RSV RGCC lineages are indicated as colored highlights over the isolate identifier and further annotated with a red square, while the G_Clade classifications are indicated as a color strip above the isolate identifier and further annotated with a red circle. Collection years 2022 and 2023 are indicated with light red and red texts, respectively, while all other collection years are indicated with black text. The Canadian isolates were primarily collected between 2022 and 2023 from four provinces, including BC, MB, ON, and SK, with corresponding text colors red, blue, purple, and green, respectively. Bootstrap values are annotated as light green dots on each branch of the tree. (B) A co-phylogeny comparing differences in clustering using the whole genome versus only the corresponding *G* gene sequence. The corresponding RGCC lineage is annotated at each tip for both phylogenetic trees.

## Characterization of aa substitutions identified among the Canadian RSVA and RSVB sequences

Among the 52 Canadian RSVA isolates sequenced in this study, a total of 271 unique aa substitutions were observed across all proteins (NS1 = 4, NS2 = 4, N = 9, P = 11, M = 6, SH = 4, G = 100, F = 20, M2−1 = 14, M2−2 = 13, and L = 86) ranging in frequency from 1.9% to 100.0% relative to hRSV/A/England/397/2017. Notable mutations flagged by RSVsurver include F:T122A (*n* = 29) and F:T122N (*n* = 2), which negate potential glycosylation of residue 120 (magenta mutations, ), as well as F:K272M (*n* = 1; RV00295, CA-SK 2016, A.D) and F:S276N (*n* = 5; RV00301-00302, RV00305, RV00316, and RV00326) (Fig. 5A). Likewise, for the 37 Canadian RSVB isolates sequenced here, a total of 228 unique aa substitutions were observed across all proteins (NS1 = 3, NS2 = 8, N = 8, P = 6, M = 22, SH = 4, G = 62, F = 30, M2−1 = 4, M2−2 = 10, and L = 71) ranging in frequency from 2.7% to 100.0% relative to hRSV/B/Australia/VIC-RCH056/2019. RSVsurver flagged the presence of F:R191K in two isolates, RV00297 and RV00298 (CA-SK 2016, B.D.4.1) (orange mutation, ), which was shown to be among residues capable of modulating RSV fusion activity *in vitro* (40).

For the Canadian RSVA isolates, no variability was observed within the aa residues corresponding to the nirsevimab targeted antigenic site Ø (62–96 and 195–227), while

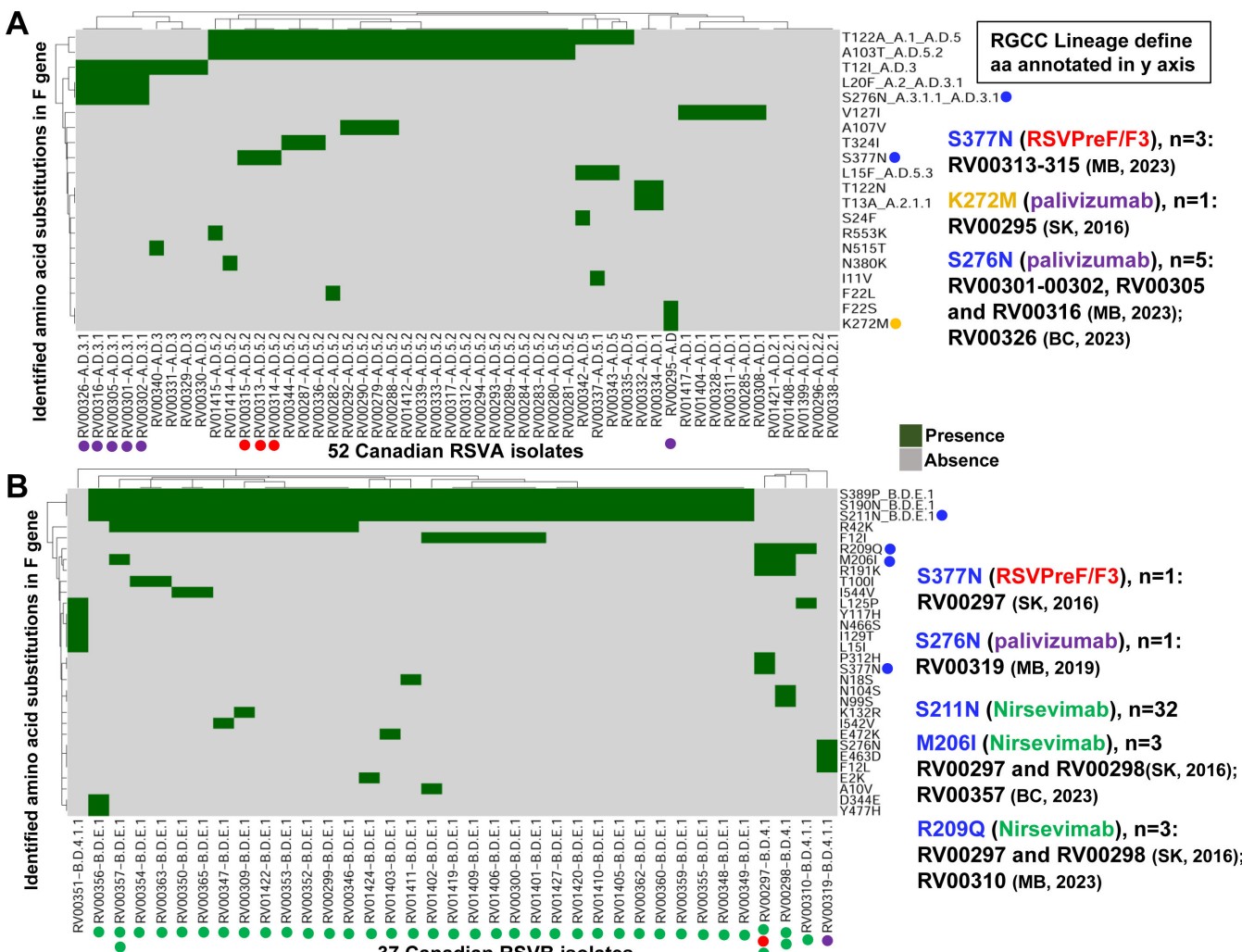

**FIG 5** A heatmap describing the distribution of amino acid substitutions identified in the *F* gene among the (A) 52 Canadian RSVA and (B) 37 Canadian RSVB isolates. The dark green and light gray colors represent the presence or absence of a particular substitution. The isolate identifier and lineage classification corresponding to each isolate are listed on the *x*-axis, while the amino acid substitutions identified in this study are listed on the *y*-axis. Substitutions written in blue font and annotated with blue circles on the *y*-axis correspond to mutations assigned as interest level 1 (moderately significant) by RSVsurver, and similarly, those written in orange font and annotated with orange circles on the *y*-axis correspond to mutations assigned as interest level 2 (significant, known to be involved in drug binding). Isolates with substitutions in the antigenic sites corresponding to the RSVpreF/F3 vaccines and/or the mAbs (palivizumab and nirsevimab) are annotated on the *x*-axis with red, purple, and green circles, respectively. Lastly, some substitutions on the *y*-axis are further annotated with RGCC lineages as they are defining features of that lineage.

variability was observed at two residues corresponding to the palivizumab targeted antigenic site II (254–277), including F:K272M (*n* = 1), as flagged by RSVsurver, as well as F:S276N (*n* = 5). Variability was also observed in one residue, F:S377N (*n* = 3), which is within the RSVPreF3 targeted antigenic site III (45–54, 301–311, 345–352, and 367–378). For RSVB, 35 out of 37 (94.6%) isolates showed variability within the aa residues corresponding to the nirsevimab targeted antigenic site Ø including F:M206I (*n* = 3), F:R209Q (*n* = 3), and F:S211N (*n* = 32), while F:S276N (*n* = 1) represented the only variability observed within antigenic site II, and F:S377N (*n* = 1) was the only change observed within antigenic site III (Fig. 5; Fig. S1 and S2).

## DISCUSSION

The recent licensing of two new RSV vaccines and an additional monoclonal antibody for use in Canada and abroad necessitates implementation of a more robust surveillance system to monitor the evolution of this pathogen. To that end, we and others have recognized the utility of the multiplex tiling PCR approach and developed assays for both RSVA and RSVB that can be used to quickly generate near-complete genome sequences from clinical specimens (30–32, 41, 42). Enrichment by PCR is less expensive and complex, faster, and more portable relative to other methods, such as capture probe-based assays, or using techniques like ultracentrifugation and host RNA depletion in combination with brute-force metagenomic sequencing, which is attractive for outbreak investigations, surveillance programs, and use in clinical settings. However, the primer design and subsequent multiplex PCR optimization can be challenging, especially for more diverse viral species, and primers may need to be updated over time as the targeted virus evolves (30). With the continued democratization of next-generation sequencing and the development of more accessible computational tools for assay design (i.e., PrimalScheme), multiplex tiling PCR assays are increasingly being developed and deployed to support outbreak investigation and response (i.e., Zika), as well as for large-scale genomic epidemiology initiatives (i.e., severe acute respiratory syndrome coronavirus 2) (30, 43).

Here, we conducted a small pilot study to test our assays against RSV-positive clinical specimens collected from four different provinces in Canada between 2016 and 2023. A total of 52 RSVA and 37 RSVB near-complete genomes were recovered and subjected to downstream phylogenetic and comparative genomic analyses. To help contextualize the Canadian sequences within the overall population structure of RSV, we downloaded a collection of lineage exemplar reference sequences and generated phylogenetic trees based on the whole genome and the *G* gene coding sequences (File S7). Nextclade was used to assign both the new RGCC lineage and G_Clade designations for each sequence, and these data were overlaid onto the phylogenetic trees (Fig. 1 to 4). We observed that the overall structures of the phylogenetic trees derived from the WGS-based and G sequences were largely congruent, though the former was able to distinguish more closely related isolates, given the presence of additional sequence data available for interrogation (Fig. 1, 2A to 4A). This highlights the sensitivity and utility of the whole genome approach to support high-resolution outbreak and trace-back investigations. Given the contemporaneous nature of the specimens used in this study, all of the RSVA and RSVB sequences contained the A.D and B.D lineage-defining *G* gene sequence duplications, respectively. Lineage A.D contains a 72 nt *G* gene duplication that emerged in 2011, and by 2017, its descendants had replaced all other lineages (15). Similarly, lineage B.D, first detected in 1999, contains a 60 nt *G*-gene duplication, and its descendants replaced all other lineages by 2009 (Fig. S3). The evolutionary impact of the duplications in these lineages is not well understood.

From a public health perspective, RSV genomes from different regions and time points provide important information on genetic changes that may affect viral pathogenicity, antigenicity, and vaccine efficacy. This knowledge can ultimately aid in the development of more effective vaccines and antiviral therapies. WGS provides a more complete understanding of the genetic diversity of RSV and permits a comprehensive genomic analysis of specific genomic regions associated with virulence, antigenicity, and drug resistance, which is important, given the fact that RSV does not employ a proofreading mechanism during replication (16). The assays described here generate near-complete genomes that can be readily characterized using both internal and publicly available tools, such as Nextclade and RSVsurver, to facilitate identification of emerging lineages, as well as the presence of biologically important or novel mutations (File S7). RSVsurver was used to visualize the aa changes identified for RSVA and RSVB (colored balls) among all available antigenic sites within the three-dimensional prefusion and postfusion structures of the RSV F glycoprotein in complex with AM22 (magenta) and Infant Antibody AD-19425 (green), respectively (Fig. S1 and S2). The prefusion

conformation possesses all six major antigenic sites (Ø, I, II, III, IV, and V), of which only I, II, III, and IV are present in the postfusion conformation (44). Analysis using RSVsurver revealed that among the Canadian RSVA isolates, one isolate from SK collected in 2016 and three isolates collected from MB in 2023 possess the F:S377N mutation located in antigenic site III targeted by the RSVPreF3 vaccine, and this mutation might be in immunodominant sites for both RSVA and RSVB isolates (45). Moreover, one RSVA isolate collected from SK in 2016 contained the F:K272M mutation located in antigenic site II, which has been shown to impact the efficacy of palivizumab (46). Similarly, among the Canadian RSVB isolates, three mutations were identified, including F:S211N ($n = 32$), F:M206I ($n = 3$), and F:R209Q ($n = 3$) that are located in antigenic site Ø targeted by nirsevimab (Fig. 5; File S7). Thus, these isolates present an opportunity to conduct downstream antigenicity and antiviral testing to study the effect of both the well-characterized and rarer mutations observed across the data set.

In conclusion, the multiplex tiling PCR assays described in this study represent a convenient and inexpensive method for the rapid generation of near-complete RSV genomes from clinical specimens. These enhanced WGS methods can ultimately contribute to the advancement of RSV research, particularly in the areas of sequencing, diagnosis, and genomic surveillance. We have also demonstrated that the sequence data generated using our assays can be readily analyzed using downstream tools, including Nextclade for lineage assignment and RSVsurver for screening of important mutations, and can also be used to support epidemiological investigations within a genomic framework. These assays and subsequent genomic analyses offer potential for serving large-scale RSV genomic surveillance with enhanced efficiency and sensitivity, which will allow researchers to better monitor genomic variability in RSV and inform public health strategies for the development and usage of vaccines and antivirals.

## ACKNOWLEDGMENTS

This study was funded by the Public Health Agency of Canada.

We thank Cole Slater from IRVC NMLB for performing all the wet lab work in this study; Darian Hole from the Computational Operational Genomics Section at NMLB, PHAC, for developing the viralassembly pipeline (https://github.com/phac-nml/viralassembly); and Ben Hetman for his assistance in adapting the R code used to generate the co-phylogeny.

## AUTHOR AFFILIATIONS

[1]National Microbiology Laboratory, Public Health Agency of Canada, Winnipeg, Manitoba, Canada
[2]Virus Detection, Cadham Provincial Laboratory, Winnipeg, Manitoba, Canada
[3]Kingston Health Sciences Centre and Queen's University, Kingston, Ontario, Canada
[4]British Columbia Centre for Disease Control, Provincial Health Services Authority, Vancouver, British Columbia, Canada
[5]Laboratory Medicine, Saskatchewan Health Authority, Regina, Saskatchewan, Canada

## AUTHOR ORCIDs

Ruimin Gao  http://orcid.org/0009-0009-1249-6756
Cody Buchanan  http://orcid.org/0000-0001-7258-7275
Kerry Dust  http://orcid.org/0000-0003-3397-9801
Nathalie Bastien  http://orcid.org/0000-0002-6126-6238

## AUTHOR CONTRIBUTIONS

Ruimin Gao, Conceptualization, Data curation, Formal analysis, Funding acquisition, Investigation, Methodology, Project administration, Resources, Software, Supervision, Validation, Visualization, Writing – original draft, Writing – review and editing | Cody Buchanan, Data curation, Formal analysis, Investigation, Methodology, Software,

Validation, Visualization, Writing – original draft, Writing – review and editing | Kerry Dust, Data curation, Methodology, Validation, Writing – review and editing | Paul Van Caeseele, Data curation, Resources, Validation, Writing – review and editing | Henry Wong, Data curation, Methodology, Resources, Validation, Writing – review and editing | Calvin Sjaarda, Methodology, Resources, Validation, Writing – review and editing | Prameet M. Sheth, Data curation, Methodology, Resources, Validation, Writing – review and editing | Agatha N. Jassem, Data curation, Methodology, Resources, Validation, Writing – review and editing | Jessica Minion, Data curation, Methodology, Resources, Validation, Writing – review and editing | Nathalie Bastien, Conceptualization, Funding acquisition, Investigation, Project administration, Resources, Supervision, Validation, Writing – review and editing

## DATA AVAILABILITY

All the complete respiratory syncytial virus genome sequences were deposited in the GISAID database, with the accession number being listed in File S7. The raw data are available under BioProject ID PRJNA1257940.

## ADDITIONAL FILES

The following material is available online.

### Supplemental Material

**File S1 (Spectrum03142-24-s0001.txt).** hRSVA-tiling reference fasta.
**File S2 (Spectrum03142-24-s0002.txt).** hRSVA-tiling reference fasta fai.
**File S3 (Spectrum03142-24-s0003.txt).** hRSVA-Tiling scheme bed file.
**File S4 (Spectrum03142-24-s0004.txt).** hRSVB-tiling reference fasta.
**File S5 (Spectrum03142-24-s0005.txt).** hRSVB-tiling reference fasta fai.
**File S6 (Spectrum03142-24-s0006.txt).** RSVB-Tiling scheme bed file.
**File S7 (Spectrum03142-24-s0007.xlsx).** Metadata corresponding to the 176 RSVA and 123 RSVB genomes analyzed in this study.
**File S8 (Spectrum03142-24-s0008.txt).** R code used to generate the heatmap and co-phylogeny figures.
**Supplemental figures (Spectrum03142-24-s0009.pdf).** Fig. S1 to S3.

### Open Peer Review

**PEER REVIEW HISTORY (review-history.pdf).** An accounting of the reviewer comments and feedback.

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
