## [Reviewer comments · Microbiology Spectrum]

Microbiology Spectrum

Whole Genome Sequencing and Phylogenetic Classification Accelerate the Implementation of Respiratory Syncytial Virus Genomic surveillance in Canada as a Pilot Study

Ruimin Gao, Cody Buchanan, Kerry Dust, Paul Van Caesele, Henry Wong, Calvin Sjaarda, Prameet Sheth, Agatha Agatha, Jessica Minion, and Nathalie Bastien

Corresponding Author(s): Ruimin Gao, Public Health Agency of Canada Ontario Manitoba Saskatchewan Regional Office

Review Timeline:

Submission Date:	December 2, 2024
Editorial Decision:	April 22, 2025
Revision Received:	May 7, 2025
Accepted:	June 29, 2025

Editor: Chuan Lim

Reviewer(s): The reviewers have opted to remain anonymous.

Transaction Report:

DOI: <https://doi.org/10.1128/spectrum.03142-24>

Re: Spectrum03142-24 (Tiled PCR amplification-based Whole Genome Sequencing and Phylogenetic Classification Accelerate the Implementation of Respiratory Syncytial Virus Genomic surveillance in Canada as a Pilot Study)

Dear Dr. Ruimin Gao:

Thank you for the privilege of reviewing your work. Below you will find my comments, instructions from the Spectrum editorial office, and the reviewer comments.

Revision Guidelines

Sincerely,
Chuan Lim
Editor
Microbiology Spectrum

Reviewer #1 (Comments for the Author):

This is a very well written paper. This study is crucial for advancing RSV genomic surveillance by introducing a high-throughput, high-sensitivity whole genome sequencing (WGS) approach that enables more effective monitoring of viral genetic diversity and evolution. The methods and results have been explained in detail in such a way that anyone can replicate their findings to perform multiplex PCR. , overall, these findings support more informed public health strategies, guiding vaccine development,

antiviral therapies, and outbreak preparedness efforts.

Reviewer #3 (Comments for the Author):

In this article by Gao et al., the research team leverages the Primalscheme approach to perform whole genome sequencing (WGS) of respiratory syncytial virus (RSV) in Canada. Capturing WGS data from RSV is essential to monitor epidemiologic impacts, including the impacts in vaccination strategies. Though this paper does capture a number of viral genomes, the enthusiasm/impact is somewhat dampened by the fact that this is not a novel approach as others have deployed Primalscheme to develop panels of primers for WGS of RSV (both RSV-A and RSV-B). The following are comments and suggestions for enhancing the manuscript:

Major:

Some of the co-authors recently published another manuscript emphasizing the development of a Primalscheme approach to sequence other Canadian RSV isolates. These papers do appear to be sequencing different isolates and perform different bioinformatic analyses on the datasets. However, I am hesitant for both papers to be emphasizing the development of this assay as part of the results. If the assay was developed in the other paper, I am unsure the development data should be presented/emphasized here again as a second manuscript. I believe it is acceptable to showcase this as a second paper that sequences more clinical samples but not as the actual development since the development data appears to already have been published. <https://www.sciencedirect.com/science/article/pii/S1386653224001215>

Minor:

- Line 106: references are provided for other sequencing methods except hybrid capture sequencing. Would be nice to just add 1-2 references for this method and RSV.
- Line 106/Methods: It was unclear why new Primalscheme primers were designed for this experiment rather than leveraging already validated sets. Examples include: <https://virological.org/t/preliminary-results-from-two-novel-artic-style-amplicon-based-sequencing-approaches-for-rsv-a-and-rsv-b/918> and <https://journals.asm.org/doi/10.1128/spectrum.03067-23>
- Line 136-137, 161-163: Are these bioinformatic tools using default settings? Please state
- Methods: Characterisation of RSVA and RSVB Sequences using Nextclade and Co-phylogeny analysis both use custom R scripts. Would recommend making these scripts publicly available through Github or another repository for reproducibility.
- Data availability: genome sequences are deposited in GISAID but raw sequencing data is not publicly available. Please upload the raw sequencing data on an appropriate ASM approved platform and provide the accession number.
- Line 303: Please provide citation for Goya et al.
- Figure 1: What do the stars represent? What is the larger red circle and square in the phylogeny? Similar with other figures with phylogenies
- Line 341-342: please provide Goya et al 2020 citation
- Figure 5 is extremely difficult to read. Even when magnified the text isn't clear. In my opinion, the figure itself is generally difficult to interpret. Rather than listing isolates, perhaps the isolates could be highlighted with a different text color for the different mutations. Also, more a personal preference aspect but the presence/absence colors are a bit jarring together.
- Discussion: Despite seeing 1-2 citations of RSV and sequencing, it appears that the way the article is framed is that this is the first case of Primalscheme for RSV, which is not the case. I would find it pertinent to acknowledge that others have deployed Primalscheme for RSV sequencing.
- References are in a different text font

May 2, 2025

Editorial team
Microbiology Spectrum
American Society for Microbiology

RE: Tiled PCR amplification-based Whole Genome Sequencing and Phylogenetic Classification Accelerate the Implementation of Respiratory Syncytial Virus Genomic surveillance in Canada as a Pilot Study

Dear Editorial team,

Thank you for reviewer comments on the above-named manuscript, submitted for consideration as a **Research Article** by ASM Microbiology Spectrum. We wish to thank the reviewers for their thoughtful and insightful assessments regarding our manuscript. We have addressed Reviewer Comments Point-by-point as indicated below:

Reviewer #1 (Comments for the Author):

- This is a very well written paper. This study is crucial for advancing RSV genomic surveillance by introducing a high-throughput, high-sensitivity whole genome sequencing (WGS) approach that enables more effective monitoring of viral genetic diversity and evolution. The methods and results have been explained in detail in such a way that anyone can replicate their findings to perform multiplex PCR. Overall, these findings support more informed public health strategies, guiding vaccine development, antiviral therapies, and outbreak preparedness efforts.*

RESPONSE: The authors greatly appreciate the reviewer's kind comments.

Reviewer #3 (Comments for the Author):

In this article by Gao et al., the research team leverages the Primalscheme approach to perform whole genome sequencing (WGS) of respiratory syncytial virus (RSV) in Canada. Capturing WGS data from RSV is essential to monitor epidemiologic impacts, including the impacts in vaccination strategies. Though this paper does capture a number of viral genomes, the enthusiasm/impact is somewhat dampened by the fact that this is not a novel approach as others have deployed Primalscheme to develop panels of primers for WGS of RSV (both RSV-A and RSV-B). The following are comments and suggestions for enhancing the manuscript:

Major:

2. *Some of the co-authors recently published another manuscript emphasizing the development of a Primalscheme approach to sequence other Canadian RSV isolates. These papers do appear to be sequencing different isolates and perform different bioinformatic analyses on the datasets. However, I am hesitant for both papers to be emphasizing the development of this assay as part of the results. If the assay was developed in the other paper, I am unsure the development data should be presented/emphasized here again as a second manuscript. I believe it is acceptable to showcase this as a second paper that sequences more clinical samples but not as the actual development since the development data appears to already have been published. <https://www.sciencedirect.com/science/article/pii/S1386653224001215>*

RESPONSE: The reviewers' insightful comments are greatly appreciated. We agree that this study is not the first to use a tiled-PCR amplification approach for RSV WGS. The manuscript was updated to reflect recent developments in the field and give due credit. The development of the assay was initiated in early 2023 which predates the citations provided by the reviewer with the exception of the assay described in the Virology blog post (<https://virology.ws/>). However, we would like to clarify that the assays described in this manuscript **are distinct and were developed independently** of the assay described in the reviewer's mentioned Wong *et al.*, 2025 paper (publically online in Feb 2025). We became aware of each other's projects at a late stage when we were soliciting our provincial and territorial (P/T) partners within the Canadian Public Health Laboratory Network for additional clinical specimens to facilitate our optimisation process and to support a more robust validation. Given that our laboratory independently developed the assays with certain modifications based on Primalscheme approach, we feel it is still appropriate to allot some level of details towards the development and optimisation process for our assays, and how the tiled-PCR amplification method compares with other whole genome sequencing approaches. Additionally, despite the similar approach used here and in the Wong *et al.*, 2025 paper, in a subset of specimens sequenced using both assays, we observed that our assay consistently recovered the complete coding region of the *L* gene. Given the Primalscheme default setting typically does not generate primers matching the ends of a genome, we supplemented the assay with **manually-designed primers** to ensure that we could recover the complete coding region of all genes ensuring the generation of high-quality genomes suitable for all downstream analyses. More clarifications were added in the revised ms version page 7, lines 151-153, page 8, lines 170-171, page 18, lines 398-400.

Thus, as the reviewer also mentioned, Wong *et al.* 2025 paper focuses on more the development and validation of the assay, while at the beginning, ours are a bit more in

description of some level of detail towards the development and optimisation process for the primers, apart from the Primalscheme default setting. Subsequently, our major focus is more on the implementation of this assay as a pilot study in Canada to support public health strategies. To better reflect this, we have removed the “tiled-PCR amplification-based” from the title in the revised ms in page 1, line 1.

Minor:

3. *Line 106: references are provided for other sequencing methods except hybrid capture sequencing. Would be nice to just add 1-2 references for this method and RSV.*

RESPONSE: Agreed. Two hybrid capture RSV sequencing were cited in the revised version in page 5, line 104, shown as below:

27) Holland LA, Holland SC, Smith MF, Leonard VR, Murugan V, Nordstrom L, Mulrow M, Salgado R, White M, Lim ES. 2023. Genomic sequencing surveillance to identify Respiratory Syncytial Virus mutations, Arizona, USA. *Emerg Infect Dis* 29:2380-2382.

28) Goya S, Sereewit J, Pfallmer D, Nguyen TV, Bakhash S, Sobolik EB, Greninger AL. 2023. Genomic characterization of respiratory syncytial virus during 2022-23 outbreak, Washington, USA. *Emerg Infect Dis* 29:865-868.

4. *Line 106/Methods: It was unclear why new Primalscheme primers were designed for this experiment rather than leveraging already validated sets. Examples include: <https://virological.org/t/preliminary-results-from-two-novel-artic-style-amplicon-based-sequencing-approaches-for-rsv-a-and-rsv-b/918> and <https://journals.asm.org/doi/10.1128/spectrum.03067-23>*

RESPONSE: As mentioned in response to Comment #2, though development of our assays commenced in early 2023, which predated the availability of the referenced publications that was missed in our initial literature review. By the time the referenced 2024 and 2025 paper were publicly available, our lab had already spent considerable time and efforts to optimise, validate and share our assays with interested Provincial/Territorial partners, and our own manuscript was well underway and undergoing internal review and revisions. Moreover, developing our own assays provided invaluable experience and helped to inform a general framework and design decision tree that can be applied towards the development of future assays should the need arise. Our used reference sequence (fasta format) and primer coordinate file (bed format) used to create the assay are now provided as Supplemental File 1, indicated in page 10, line 226-227.

5. *Line 136-137, 161-163: Are these bioinformatic tools using default settings? Please state*

RESPONSE: Yes, default settings were used unless specified within the text. For example, in page 7, lines 139-141, the default parameters were used for Primalscheme except for the amplicon size settings. Similarly, in page 8, lines 165-167, default parameters were used for Primalscheme, except for the amplicon size settings.

6. *Methods: Characterisation of RSVA and RSVB Sequences using Nextclade and Co-phylogeny analysis both use custom R scripts. Would recommend making these scripts publicly available through Github or another repository for reproducibility.*

RESPONSE: For the sake of clarity, there was no custom R script associated with the Nextclade analysis. However, the R code used to generate the co-phylogeny and heatmaps has now been made publicly available here as a Supplemental File 3. The added description can be found in page 12, lines 255-256, and page 13, line 279-280.

7. *Data availability: genome sequences are deposited in GISAID but raw sequencing data is not publicly available. Please upload the raw sequencing data on an appropriate ASM approved platform and provide the accession number.*

RESPONSE: Thanks for the reminder. The human dehosted sequencing raw data fastq files are now deposited in the ASM approved Genebank SRA <https://submit.ncbi.nlm.nih.gov/subs/sra/>, and its BioProject ID is **PRJNA1257940**, with the access link <http://www.ncbi.nlm.nih.gov/bioproject/1257940>. This information was also added in the revised manuscript page 23, lines 508-509.

8. *Line 303: Please provide citation for Goya et al.*

RESPONSE: The mentioned citation is now shown in page 14, line 313.

9. *Figure 1: What do the stars represent? What is the larger red circle and square in the phylogeny? Similar with other figures with phylogenies*

RESPONSE: We apologize for the confusion caused. The blue stars are meant to draw the reader's attention to the fact that sequences from the two oldest Canadian specimens in the study, collected in 2016 from Saskatchewan, cluster with other contemporaneous sequences (Figure 1; text description in yellow highlighted in pages 14-15, lines 316-318).

Similarly, in page 15, lines 333-338, the green stars are meant to demonstrate an example of how WGS can resolve differences in closely related sequences as compared to using the G gene sequence. For example, within the A.D.5.2 lineage, four isolates, RV00279 (CA-SK 2023), RV00288 (CA-SK 2023), RV00290 (CA-SK 2023) and RV00292 (CA-SK 2023) were indistinguishable using the G gene sequence (green stars; Figure 2A), but at the WGS level, could be resolved into two pairs of identical sequences: RV00279 (CA-SK 2023) and RV00288 (CA-SK 2023), as well as RV00290 (CA-SK 2023) and RV00292 (CA-SK 2023) (green stars, Figure 1). The large red circle corresponds to the corresponding RGCC lineages designation, and large red square corresponds to the corresponding G_Clade designation. To remedy this, we have better annotated the new figures 1-4, and improved the figure legends in page 28, lines 683-685, lines 694-697; page 29, lines 706-708, lines 716-719.

10. Line 341-342: please provide Goya et al 2020 citation

RESPONSE: The citation has been added, and now can be found in page 16, lines 342-343.

11. Figure 5 is extremely difficult to read. Even when magnified the text isn't clear. In my opinion, the figure itself is generally difficult to interpret. Rather than listing isolates, perhaps the isolates could be highlighted with a different text color for the different mutations. Also, more a personal preference aspect but the presence/absence colors are a bit jarring together.

RESPONSE: We apologize for the legibility of the original version of Figure 5. To remedy this, we revised and re-draw Figure 5 to include only the Canadian RSV A (n=52) and RSV B (n=37) isolates used in this study. We hope that the new green/light grey colour scheme is more appealing and easier to view. One more sentence was added for RSV B in page 18, lines 390-391. The Figure 5 legend was revised accordingly to reflect all the changes in page 30, lines 726-737.

12. Discussion: Despite seeing 1-2 citations of RSV and sequencing, it appears that the way the article is framed is that this is the first case of Primalscheme for RSV, which is not the case. I would find it pertinent to acknowledge that others have deployed Primalscheme for RSV sequencing.

RESPONSE: Thank you for this comment. As mentioned in other responses, we commenced work on this project in early 2023 and the initial drafts of the manuscript were written prior to publication of the papers referenced by the reviewer. However, we wholeheartedly agree with correcting the language in the manuscript to reflect the current

state of affairs and to ensure proper credit is given. More recent paper were also cited in the revised version in **page 18, line 400**. Throughout the whole ms, we have revised: **pages 1-2, line 25-27, lines 37-38; page 5, lines 106-112; page 18, lines 398-400**.

13. References are in a different text font

RESPONSE: References text font was changed in **pages 24-28**, which is now consistent with the main text.

If you have any questions, or require additional information, please do not hesitate to let us know.

Thank you again for your consideration.

Ruimin Gao & Nathalie Bastien

National Microbiology Laboratory
Public Health Agency of Canada

Re: Spectrum03142-24R1 (**Whole Genome Sequencing and Phylogenetic Classification Accelerate the Implementation of Respiratory Syncytial Virus Genomic surveillance in Canada as a Pilot Study**)

Dear Dr. Ruimin Gao:

Your manuscript has been accepted, and I am forwarding it to the ASM production staff for publication. Your paper will first be checked to make sure all elements meet the technical requirements. ASM staff will contact you if anything needs to be revised before copyediting and production can begin. Otherwise, you will be notified when your proofs are ready to be viewed.

Sincerely,
Chuan Lim
Editor
Microbiology Spectrum

Reviewer #1 (Comments for the Author):

N/A

Reviewer #3 (Comments for the Author):

Thank you to the authors for providing responsive, clear, and informative responses. I have no further comments or concerns. A well-done article!

The authors have addressed the comments and suggestions from the initial round of review thoughtfully and comprehensively. The revised manuscript maintains its strong contribution to RSV genomic surveillance by demonstrating a robust and scalable WGS approach with high sensitivity. The added clarifications in the methodology and improved organization of the results section further enhance the reproducibility and clarity of the study. Overall, the revisions have strengthened the manuscript, and I believe it is now suitable for publication. This work will be of significant value to the field of viral genomics and public health preparedness.